# MentaLMs: Teaching Language Models to Reason Efficiently in the Language of Thought

## Abstract

Large Reasoning Models (LRMs) achieve state-of-the-art performance in mathematics, code generation, and task planning. However, their reliance on long chains of verbose "thinking" tokens results in high latency, redundancy, and incoherent reasoning paths. Inspired by the Language of Thought Hypothesis —which posits that human reasoning operates over a symbolic, compositional mental language called *Mentalese*—we introduce a cognitively motivated framework that trains models to reason in a similar compact style, which we call MentaLMs. *Mentalese* encodes abstract reasoning as ultra-compressed, structured tokens, enabling models to solve complex problems with far fewer steps. To achieve both efficiency and accuracy, we propose Shorter Length Preference Optimization (SLPO), a reinforcement learning method that directly optimizes models to generate concise yet correct reasoning by rewarding shorter solutions that maintain high accuracy while flexibly allowing longer reasoning when complexity demands it. When applied to *Mentalese*-aligned models, SLPO achieves much larger compression rates by enabling compressed reasoning that preserves the benefits of detailed thinking without the computational overhead, allowing us to present the best-performing models at each compression level along the performance-efficiency Pareto frontier. Across mathematical benchmarks—including AIME 2024, Minerva, OlympiadBench, Math500, and AMC—our MentaLMs generates reasoning traces with 4–16× fewer tokens, achieves over 5× lower inference latency, and reduces training costs by 7–9× relative to the base DeepSeek R1 Distilled model, while maintaining 90-98% of baseline accuracy. MentaLMs also surpasses Claude and ChatGPT-4o by up to 5% in accuracy while maintaining 2× compression. Our findings demonstrate *Mentalese*-style compressed reasoning offers a breakthrough toward human-like cognitive efficiency, opening new possibilities for real-time, cost-effective reasoning without sacrificing accuracy.

## 1 Introduction

Recent advances such as OpenAI o1 OpenAI et al. (2024b) and DeepSeek R1 (DeepSeek-AI et al., 2025) have reshaped how we think about language model reasoning. By letting models "think before they answer," these systems dramatically improved credibility and performance—achievements that were once thought impossible for LLMs (Wu et al., 2024). Explicit reasoning has thus emerged as a central focus in LLM research (Xu et al., 2025). Recent work such as DeepScaleR: Surpassing O1-Preview with a 1.5B Model by Scaling RL demonstrates that even relatively small models (1.5B parameters) can outperform OpenAI's O1-Preview—which is widely assumed to be significantly larger, though its scale has not been publicly disclosed—by leveraging reasoning-focused reinforcement learning techniques such as RLVR, where models generate intermediate "thinking" tokens for self-verification (Luo et al., 2025b). This finding underscores that scaling in reasoning depth can, in some contexts, rival scaling in parameter size. The key challenge now lies in transforming this promise into robust, efficient, and trustworthy deployments, which we address in the next section. However, the promise of RLVR comes with significant trade-offs. Training is computationally expensive, with rollout generation leaving GPUs idle for long periods (Fu et al., 2025). Even relatively small models such as 1.5B parameters can take days to train under RL fine-tuning regimes (Zheng et al., 2025). Moreover, R1-style reasoning traces (shown in Figure 1a) are often verbose, redundant, and unnatural - a far cry from human cognition, which tends to rely on short and efficient thought steps (Sui et al., 2025). Building on the Language of Thought hypothesis (Fodor, 1975), which sug-

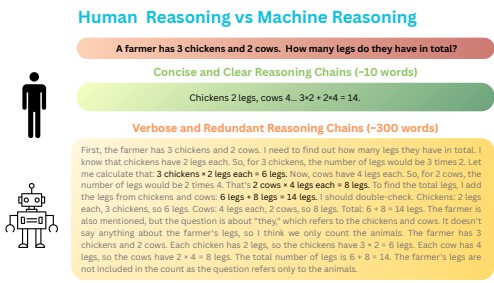

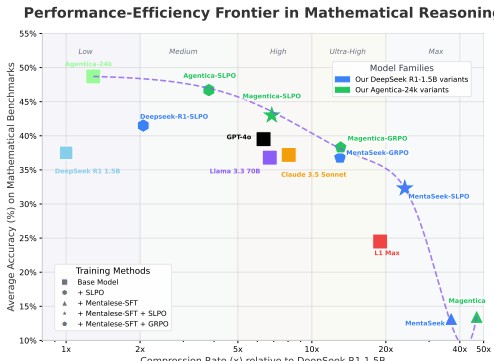

(a) Contrast between human and machine reasoning (response from Deepseek-R1). While humans arrive at intuitive and concise solutions, LLMs often produce verbose and redundant reasoning chains even for simple problems. We bridge this gap by developing methods that encourage models to reason more like humans—clear, efficient, and direct—while preserving accuracy. Grounded in the Language of Thought hypothesis, human reasoning compresses complex ideas into minimal symbolic steps, reflecting cognitive efficiency. Emulating this compact reasoning reduce redundancy in machine outputs, improving both interpretability and token efficiency.

(b) Performance–efficiency trade-offs of various model families across six mathematical reasoning benchmarks (including AIME2025). The dotted curve indicates the Pareto frontier, which illustrates the trade-off between higher compression rates and loss in accuracy. Our proposed method, combining Mentalese with SLPO, consistently lies on this frontier, identifying an optimal operating point that achieves a balance between accuracy and efficiency.

Figure 1: Contrast between human and machine reasoning approaches. While humans use intuitive and concise steps, LLMs often produce verbose and redundant reasoning for even simple problems.

gests that human cognition unfolds through short compositional thought units rather than verbose natural language traces, we propose a training framework that restructures the reasoning style of current reasoning-oriented LLMs in a symbolic internal language that we call *Mentalese*.

In our framework, models are first aligned with this reasoning process through supervised fine-tuning on reasoning traces in *Mentalese*, namely, concise compositional sequences that capture only the essential steps required for problem solving. However, aligning models to *Mentalese* by supervised fine-tuning alone leads to a substantial drop in performance relative to the base model. To overcome this, we introduce *Shorter Length Preference Optimization* (SLPO), a reinforcement learning objective with a verifiable reward that balances brevity and correctness. Unlike token-penalization methods (e.g., L1-style objectives (Aggarwal & Welleck, 2025)) that impose arbitrary length budgets—often forcing models to under-reason on difficult problems and over-reason on easy ones—SLPO instead rewards relative efficiency: among correct rollouts, shorter solutions receive a bonus. This naturally biases the model toward concise reasoning when tasks are simple, while still allowing it to allocate more steps when necessary in the same reasoning structure. By rewarding concise but correct solutions, SLPO avoids verbosity while recovering most of the performance lost during supervised fine-tuning on Mentalese, thereby yielding efficient reasoning that scales at inference time.

To highlight both domain-specific performance and generalization, we evaluated our suite of trained models (*MentaLMs 1.5B*) on mathematical reasoning in the domain and out-of-the-domain tasks such as GPQA, LSAT, and MMLU. Our *Magentica-SLPO 1.5B* surpasses GPT-4o (OpenAI et al., 2024a), Claude 3.5 Sonnet (Anthropic, 2024), and Llama 3.3 70B (Grattafiori et al., 2024) by an average of 6 pp[1] in mathematical reasoning and outperforms DeepSeek-R1 1.5B DeepSeek-AI et al. (2025) by 7 pp with a 7× reduction in reasoning length (Figure 1b). *MentaSeek-GRPO 1.5B* achieves 14× compression relative to DeepSeek-R1 1.5B. Beyond in-domain gains, our *MentaLMs* also generalize well: on out-of-domain tasks they improve over the base model by 1 pp while achieving 15× compression Table 2. In addition to token efficiency, our experiments show that training with Mentalese stabilizes optimization and reduces training time by 7–9× compared to directly training the base model with RLVR, leading to substantial savings in training cost. Beyond benchmarks, we hypothesize that these ideas are especially relevant for agentic LLM systems, where reasoning models are rarely deployed due to latency and cost: verbose generations can overwhelm communication channels Kim et al. (2025). A compressed reasoning style, reinforced through SLPO, has the poten-

---

[1]pp = percentage points, denoting absolute differences between percentages (e.g., 22% vs. 16% = 6 pp).

tial to dramatically reduce this overhead—making reasoning-capable agents not only more accurate but also more practical to deploy in real-world settings. Our main contributions are as follows.

- **Reasoning compression framework.** We propose a novel and efficient reasoning compression framework via *Mentalese* alignment for restructuring the reasoning style of current LLM, producing compact yet faithful symbolic reasoning.

- **Reward function.** We propose *Shorter Length Preference Optimization (SLPO)*, an adaptive objective that dynamically balances correctness with brevity, eliminating the need for rigid length penalties.

- **Dataset.** We release *MentaleseR-40k*, a dataset of ultra-compressed reasoning traces for 40k math problems, generated under symbolic constraints inspired by the Language of Thought Hypothesis (LOTH), to support future developments and foster research on efficient reasoning.

- **Experiments and best practices.** We conduct extensive evaluations and identify best practices to apply GRPO and SLPO, showing how they achieve different levels of compression and where each method is most effective.

## 2 RELATED WORKS

**Efficient Reasoning in Large Language Models:** Since Wei et al. (2022b) demonstrated the effectiveness of chain-of-thought (CoT) prompting, subsequent work has focused on scaling test-time computation to improve performance in mathematical problem-solving, code generation, and complex reasoning tasks. Strategies include parallel sampling of multiple reasoning paths (Wang et al., 2022; Yue et al., 2024; Chen et al., 2023), tree search and planning (Yao et al., 2023; Besta et al., 2024), and iterative refinement methods (Madaan et al., 2023). Recent reasoning-specialized models, such as OpenAI's *o1* (OpenAI et al., 2024b), DeepSeek-R1 (DeepSeek-AI et al., 2025), and Qwen-QwQ (Yang et al., 2025), internalize the ability to generate extended reasoning traces.However, these methods often suffer from the *overthinking phenomenon* (Sui et al., 2025; Chen et al., 2025), where models generate excessively long reasoning traces. While increased length can improve accuracy up to a point (Wu et al., 2025), it also introduces redundancy, higher inference latency, and even accuracy degradation due to compounding errors Hassid et al. (2025); Lee et al. (2025). This trade-off has motivated work on more efficient reasoning. RL-based post-training methods have been widely explored to control reasoning length. L1 (Aggarwal & Welleck, 2025) enforces user-specified budgets, DAST (Shen et al., 2025) adapts budgets based on problem difficulty, while O1-Pruner (Luo et al., 2025a) uses reference-based pruning. Other approaches, such as Kimi 1.5 (Team et al., 2025) and Training Efficient (Arora & Zanette, 2025), use sampled rollouts to reward shorter or average lengths. ShorterBetter (Yi et al., 2025) further introduces the idea of rewarding the shortest correct response, highlighting the existence of problem-dependent optimal reasoning lengths. Our work complements these by introducing SLPO, which adaptively prefers concise correct reasoning without penalizing necessary longer derivations, enabling over $10\times$ compression with minimal loss in accuracy.

**Chain-of-Thought and Alternative Reasoning Formats**: CoT reasoning has become a dominant paradigm for enhancing reasoning in LLMs, either via prompting Wei et al. (2022a); Fu et al. (2022); Zhou et al. (2022) or through post-training with supervised finetuning Yue et al. (2023); Yu et al. (2023) and reinforcement learning Luong et al. (2024); Shao et al. (2024b); Zhou et al. (2025). Theoretical analyses link CoT to increased expressivity and effective depth in transformers Feng et al. (2023); Merrill et al. (2023); Li et al. (2024). However, natural-language CoT traces are verbose, redundant, and not always faithful to the model's underlying reasoning process Turpin et al. (2023); Chuang et al. (2024). Recent research has explored alternatives. Structured or symbolic CoT formats aim to compress reasoning into more compact representations, such as symbolic operators, patterns, or abstract primitives Madaan & Yazdanbakhsh (2022); Paul et al. (2024). Other works examine latent reasoning, where intermediate computation is implicit in hidden representations rather than externalized tokens Yang et al. (2024); Biran et al. (2024); Shalev et al. (2024). Techniques such as back-patching Biran et al. (2024), filler tokens Pfau et al. (2024), or knowledge distillation into latent reasoning Deng et al. (2023; 2024) push beyond explicit CoT. Our proposed *Mentalese Chain-of-Thought* builds on this line of work by introducing a symbolic, cognitively motivated reasoning language inspired by the Language of Thought Hypothesis. By replacing verbose

natural language with structured symbolic primitives, Mentalese CoT achieves order-of-magnitude compression while retaining faithfulness and interpretability. Combined with SLPO, this framework demonstrates that both representation and optimization are critical for efficient and reliable reasoning.

## 3 METHODOLOGY

In this section, we present our methodology, which integrates symbolic reasoning alignment with reinforcement learning for concise yet accurate performance. We introduce *Mentalese*, a compact symbolic reasoning format, and *Group Relative Policy Optimization (GRPO)*, a group-based extension of PPO for reasoning optimization. Our main contribution, *Shorter Length Preference Optimization (SLPO)*, refines GRPO by rewarding brevity without penalizing necessary longer reasoning. Finally, we propose *RLVR*, a two-stage pipeline that first aligns models to Mentalese via supervised finetuning, then applies GRPO or SLPO with verifier feedback. Together, these components yield 10–20$\times$ compression in reasoning traces while maintaining accuracy and efficiency across benchmarks.

### 3.1 MENTALESE: MENTAL LANGUAGE OF THOUGHT

We first introduce *Mentalese*, a cognitively motivated reasoning format inspired by the Language of Thought Hypothesis (Fodor, 1975; Rescorla, 2024). According to this hypothesis, human cognition operates not directly in natural language, but in an internal representational system characterized by compact, symbolic structures. Translating this perspective to Large Reasoning Models (LRMs), we hypothesize that verbose natural language explanations commonly used in chain-of-thought prompting especially the Deepseek R1 reasoning style, are not essential for reasoning, and that more efficient symbolic primitives can better capture the core logical operations underlying problem-solving.

**Formal definition.** Let $\mathcal{O}$ be a finite set of operators (e.g., SET, EQ, CASE, SOLVE, CALC, DIFF, ANS) and let $\mathcal{E}$ be the set of symbolic expressions over variables, numbers, and function symbols (e.g., $+, -, \times, \div$, abs). A *Mentalese step* is a pair $s_t = (o_t, c_t)$ with $o_t \in \mathcal{O}$ and *expression* $c_t \in \mathcal{E}$ rendered as the string OPERATION:expression;. A *Mentalese trace* for a question $q$ is a finite sequence $M = (s_1; \ldots; s_T)$ that is well-typed and executable under the step semantics below and that culminates in exactly one terminal ANS:e; step. The boxed final answer is $e^\star$, where $e^\star$ is the value denoted by $e$. We denote the set of valid traces by $\mathcal{M}$.

Unlike traditional CoT, which uses free-form text, *Mentalese* encodes reasoning in canonical steps of the form OPERATION:expression;, joined by semicolons to form minimal yet complete traces. This yields three advantages: (i) Compression — eliminating redundant tokens for up to 10$\times$ shorter reasoning; (ii) Faithfulness — each step is necessary and sufficient; (iii) Cognitive alignment — resembling structured mental representations rather than verbose text.

To build MENTALESE-40K, we adapted the DEEPSCALER-PREVIEW-DATASET (Luo et al., 2025b), covering 40k+ math problems from AIME (1983–2023), Omni-Math, and STILL. We used GPT-4.1 with a structured prompting framework (Figure 2a)—including a formal definition, syntactic rules, and examples—to generate *Mentalese* traces. After light curation (removing 65 malformed cases), the resulting dataset was used for supervised fine-tuning. For RLVR, we instead relied on the original QA pairs, letting the verifier assess correctness while optimizing for concise reasoning. Refer Appendix A.3 for some of the samples from MENTALESER-40K

### 3.2 GROUP RELATIVE POLICY OPTIMIZATION (GRPO)

While PPO (Schulman et al., 2017) provides a strong baseline for policy optimization, it operates at the *single-sample* level: each rollout is evaluated independently using a value function to estimate its advantage. However, in reasoning tasks where multiple candidate solutions can be generated for the same question, evaluating rollouts in isolation discards useful information about the *relative quality* of responses within a group. For example, if a model generates five candidate solutions, some correct and some incorrect, we are less interested in their absolute values than in how each compares relative to others in the same set. This motivates **Group Relative Policy Optimization (GRPO)** (Shao et al., 2024a), which eliminates the explicit value function and instead estimates the advantage by normalizing rewards *within groups of samples* drawn for the same prompt. Concretely,

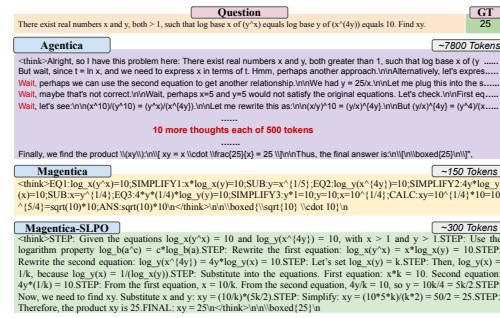

**MENTALESE FRAMEWORK**

**DEFINITION**

*A symbolic, logic-based chain of thought using compact primitives and logical steps only. Each step should be separated by a semicolon (;) with no natural language. All core logic steps must be preserved.*

**EXAMPLE**

SET:w;
EQ:abs(180-5.5*w)=110;
CASE1:180-5.5*w=110;
SOLVE1:w=70/5.5;
CASE2:180-5.5*w=-110;
SOLVE2:w=290/5.5;
CALC1:w1=70/5.5=12+8/11;
CALC2:w2=290/5.5=52+8/11;
DIFF:t=w2-w1=40;
ANS:40

**RULES**

1. Use ONLY symbolic notation in the form: OPERATION: expression

2. No natural language

3. No redundancy

4. Every step must be necessary and sufficient for solving the problem

(a) Illustration of symbolic, logic-based chain of thought (mentalese). This figure shows the definition (top), an example of symbolic reasoning steps (left) with rules governing the reasoning style (right).

(b) Comparison of reasoning traces on AIME 2024: Agentic models use 7800 tokens, Magentica 150 tokens, and Magentica-SLPO 300 tokens with similar accuracy.

Figure 2: Illustration of symbolic, logic-based chain of thought (mentalese).

for a question–answer pair $(q, a)$, we sample a group of $G$ responses $\{o_i\}_{i=1}^{G}$ from the current policy. The reward of each response $r_i$ is converted into a *group-relative advantage* via normalization:

$$\hat{A}_i = \frac{r_i - \mathrm{mean}(\{r_j\}_{j=1}^{G})}{\mathrm{std}(\{r_j\}_{j=1}^{G}) + \epsilon}.$$

This design ensures that advantages highlight which responses are better or worse relative to the group, rather than depending on an absolute critic model. GRPO then optimizes a clipped surrogate objective similar to PPO but with a directly imposed KL penalty:

$$\mathcal{J}_{\mathrm{GRPO}}(\theta) = \mathbb{E}\left[\frac{1}{G}\sum_{i=1}^{G}\frac{1}{|o_i|}\sum_{t=1}^{|o_i|}\Big(\min\big(r_{i,t}(\theta)\hat{A}_{i,t}, \mathrm{clip}(r_{i,t}(\theta), 1-\varepsilon, 1+\varepsilon)\hat{A}_{i,t}\big) - \beta D_{\mathrm{KL}}(\pi_\theta\|\pi_{\mathrm{ref}})\Big)\right],$$

where $r_{i,t}(\theta) = \frac{\pi_\theta(o_{i,t}|q,o_{i,<t})}{\pi_{\mathrm{old}}(o_{i,t}|q,o_{i,<t})}$ is the token-level importance ratio.

### 3.3 Shorter Length Preference Optimisation (SLPO)

There has been a growing interest in adaptive reasoning methods that redefine the GRPO formulation by incorporating explicit *thinking budgets*. For instance, prior works such as LCPO constrain reasoning lengths by enforcing unnatural fixed or maximum token budgets. More recently, group-relative formulations have also been proposed that define rewards based on the relative lengths of responses within a group. However, these methods tend to be overly rigid: they *over-penalize* the longest solutions even when they converge to right answer, and in cases where no correct solution exists, their length-normalization can still distort the reward landscape. This strictness can suppress necessary reasoning and lead to degenerate behavior.

To overcome these issues, we introduce **Shorter Length Preference Optimization (SLPO)**, a reinforcement learning strategy that balances conciseness with correctness *softly*. Crucially, SLPO never penalizes a correct but necessarily long reasoning when it is the only valid option, and it does not distort rewards in cases with no correct solution. Instead, it adaptively rewards shorter correct traces when multiple valid derivations exist, while preserving correctness as the primary training signal.

Different problems naturally require different amounts of reasoning. For example, a simple arithmetic task such as $2 + 2$ requires no intermediate steps, while Olympiad-style geometry problems demand much longer derivations. A reward function that ignores this variability either pushes the model toward artificially verbose chains (reward hacking under fixed budgets) or toward overly terse and often incorrect responses (under strict length penalties). SLPO resolves this by defining *preferences relative to the observed range of correct reasoning lengths* for each problem instance.

Formally, for a given rollout group $G(x_i) = \{y_1, y_2, \ldots, y_n\}$ corresponding to prompt $x_i$, let $\mathcal{C}(x_i) = \{y_j \in G(x_i) : R_{\mathrm{correctness}}(y_j) = 1\}$ denote the set of correct responses. We define:

$$L_{\min} = \min_{y\in\mathcal{C}(x_i)} \ell(y), \qquad L_{\max} = \max_{y\in\mathcal{C}(x_i)} \ell(y), \tag{1}$$

where $\ell(y)$ is the token length of response $y$. The total reward for candidate $y_{\text{curr}}$ is then:

$$
R_{\text{SLPO}}(y_{\text{curr}}) = \begin{cases} 1, & \text{if } |\mathcal{C}(x_i)| = 1 \text{ or } \big(|\mathcal{C}(x_i)| > 1 \ \& \ L_{\min} = L_{\max}\big), \\[2ex] R_{\text{correctness}} + \alpha \cdot \dfrac{L_{\max} - L_{\text{curr}}}{L_{\max} - L_{\min}}, & \text{if } |\mathcal{C}(x_i)| > 1 \text{ and } L_{\min} \neq L_{\max}, \\[2ex] 0, & \text{if } |\mathcal{C}(x_i)| = 0, \end{cases}
\tag{2}
$$

where $L_{\text{curr}}$ is the length of the current response, $R_{\text{correctness}} \in \{0,1\}$ is determined by a verifier, and $\alpha$ controls the trade-off between accuracy and conciseness. Larger values of $\alpha$ emphasize brevity, whereas smaller values prioritize correctness regardless of length. In all experiments, we set $\alpha = 0.1$, which provided a stable balance across benchmarks.

By construction, SLPO avoids the failure modes of previous group-relative and L1-based formulations: it does not over-penalize long but uniquely correct solutions, and it does not distort reward landscapes when no valid solutions exist. Instead, it consistently encourages models to discover the *shortest correct reasoning trace* whenever possible. This makes SLPO especially well-suited for mathematical reasoning, where optimal reasoning lengths vary significantly across problems.

### 3.4 MENTALESE ALIGNMENT THROUGH SFT FOLLOWED BY RLVR

We now present our complete training pipeline, which consists of two stages: supervised alignment on *Mentalese* traces, followed by reinforcement learning with verifiable rewards (RLVR).

**Stage 1: Supervised Finetuning on Mentalese.** Let $\mathcal{D} = \{(q_i, a_i, m_i)\}_{i=1}^{M}$ be our dataset with question $q_i$, ground-truth final answer $a_i$, and Mentalese reasoning trace $m_i$. Each training prompt is structured as

$\tau(q_i) = q_i +$ `'Let's think step-by-step and answer within \boxed{}.'`

with target output

$$y_i^{\star} = \texttt{<think>} \ m_i \ \texttt{</think>} \ \texttt{\textbackslash boxed}\{a_i\}.$$

Starting from a pretrained base model $\pi_0$, we obtain a Mentalese-aligned model $\pi_{\text{SFT}}$ via supervised finetuning:

$$
\pi_{\text{SFT}} = \arg\min_{\theta} \ -\frac{1}{M} \sum_{i=1}^{M} \log \pi_{\theta}(y_i^{\star} \mid \tau(q_i)).
$$

**Stage 2: Reinforcement Learning with Verifier Rewards (RLVR).** The SFT model $\pi_{\text{SFT}}$ is further refined using verifier-based reinforcement learning. For each question $q_i$, the policy generates $N$ candidates $G(q_i) = \{y^{(1)}, \dots, y^{(N)}\}$; a verifier checks the boxed answer $\hat{a}$ against $a_i$ and assigns a correctness reward $R_{\text{acc}} \in \{0,1\}$. The clipped surrogate objective with KL regularization is:

$$
\mathcal{J}_{\text{RLVR}}(\theta) = \mathbb{E}_{q \sim \mathcal{D}, \ y \sim \pi_{\theta}(\cdot|q)} \Big[ \min\big(r_{\theta}(y)\hat{A}(y), \ \text{clip}(r_{\theta}(y), 1-\varepsilon, 1+\varepsilon)\hat{A}(y)\big) - \beta D_{\text{KL}}(\pi_{\theta} \parallel \pi_{\text{SFT}}) \Big],
$$

where $r_{\theta}(y) = \frac{\pi_{\theta}(y|q)}{\pi_{\text{old}}(y|q)}$ and $\hat{A}(y)$ is computed from either the **GRPO** or **SLPO** formulation (see previous subsections).

Depending on the chosen reward function, RLVR yields a policy $\pi_{\text{GRPO}}$ or $\pi_{\text{SLPO}}$:

$$
\pi_{\text{SFT}} \xrightarrow{\text{RLVR (GRPO)}} \pi_{\text{GRPO}}, \qquad \pi_{\text{SFT}} \xrightarrow{\text{RLVR (SLPO)}} \pi_{\text{SLPO}}.
$$

SFT alignment anchors the model to a compact single-chain reasoning format, ensuring that outputs conform to the Mentalese structure shown in Figure 2b. However, this alignment often comes at the cost of reduced accuracy, since the base model initially performs well with long and verbose reasoning chains. RLVR provides the complementary step: by instantiating it with either GRPO or SLPO, the model learns to recover accuracy while retaining the compact reasoning format. RLVR enables the model to refine and extend its reasoning inside the learned structure, adding useful steps when necessary but avoiding unnecessary verbosity. This combination not only restores the lost performance but also yields consistent improvements in overall reasoning efficiency and accuracy across benchmarks.

## 4 EXPERIMENT DESIGN

Our experimental study is organized around five research questions. First, we ask how effective the proposed *Mentalese* representation is in compressing reasoning traces while preserving task performance. Second, we investigate which reinforcement learning algorithm (GRPO or SLPO) best recovers the performance gap introduced by compression. Third, we evaluate the standalone effectiveness of SLPO in balancing conciseness and correctness. Fourth, we explore best practices by identifying when each algorithm is most suitable, particularly across different regions of compression. Finally, we analyze the efficiency and stability of our RLVR methods during training.

**Models and Baselines.** To evaluate our method, we consider both base models and competitive baselines. Our primary comparison point is DeepSeek-R1-Distill-Qwen-1.5B DeepSeek-AI et al. (2025), a distilled variant of Qwen-2.5-1.5B-Instruct fine-tuned on reasoning traces from DeepSeek's R1 model, which we denote as DEEPSEEK-R1-1.5B. We also include DeepScaleR-1.5B-Preview Luo et al. (2025b), the original release without length-control modifications, referred to as AGENTICA-24K. For completeness, we tested the base Qwen-2.5-Math-1.5B model Qwen et al. (2025); however, it collapsed under RLVR fine-tuning due to NaN gradient norms, so it is excluded from final evaluations. In addition, we benchmark against L1-Max Aggarwal & Welleck (2025), a strong baseline derived from AGENTICA-24K using Length-Controlled Policy Optimization (LCPO), which achieves more than 10× compression. While this approach effectively reduces verbosity by enforcing a fixed token budget, it lacks adaptability to varying problem difficulties. Beyond these 1.5B-scale models, we also report results from frontier-scale systems for context. Specifically, we include GPT-4o OpenAI et al. (2024a), Claude 3.5 Sonnet Anthropic (2024), and LLaMA-3 70B-Instruct Grattafiori et al. (2024) as strong reference points, situating our results relative to state-of-the-art closed-source and large-scale open-source LLMs.

**Evaluation and Metrics.** We evaluate our models on five in-domain mathematical reasoning datasets: **AIME 2024** (Mathematical Association of America, 2024), **AIME 2025**(Mathematical Association of America, 2025), **MATH-500** (Hendrycks et al., 2021b), **AMC** (American Mathematics Competitions ), **Minerva-Math** Lewkowycz et al. (2022), and **OlympiadBench** He et al. (2024). Additionally, we test on three out-of-domain benchmarks: **GPQA** (Rein et al., 2023), **LSAT** (Zhong et al., 2023), and **MMLU** (Hendrycks et al., 2021a), in order to assess generalization beyond mathematical reasoning. We report results using three primary metrics. Pass@1 measures the fraction of problems correctly solved under single-sample decoding, i.e., the proportion of test questions for which the model produces a correct solution on its first attempt. Token Length denotes the average number of tokens generated per response on a given benchmark, computed by averaging output lengths across all test questions. Compression Rate (CR) quantifies the degree of response shortening relative to DeepSeek-R1-1.5B, with higher values indicating greater compression (e.g., a CR of 10 means the model's responses are ten times shorter on average). Full formal definitions are provided in Appendix A.2.

**Implementation Details.** For supervised fine-tuning on the MENTALESER-40K dataset, we used LLAMA-FACTORY Zheng et al. (2024), an open-source library for instruction tuning and post-training. For reinforcement learning, we adopted VERL Sheng et al. (2025), an open-source RL training library. We fine-tuned our 1.5B base models with a batch size of $N = 128$ and a rollout group size of $n = 16$. Training was conducted for 1500 steps with a fixed learning rate of $1 \times 10^{-6}$. For reinforcement learning experiments, we used 32 H100 GPUs, while supervised fine-tuning was performed on 8 H100 GPUs. Inference was accelerated using the vLLM Kwon et al. (2023) engine, which enables efficient large-scale generation. For length constraints, we set different maximum generation lengths depending on the training setup: 8K tokens for direct SLPO on base models, 2K tokens for SLPO on MENTALESER-40K fine-tuned models, and 1K tokens for GRPO on MENTALESER fine-tuned models. During evaluation on benchmarks, we fixed a maximum generation budget of 8K tokens. More details are provided in Appendix A.2.2.

## 5 DISCUSSION

**Loss During SFT, Recovery Through RLVR.** As shown in Table 1, we observed a substantial performance drop after the SFT stage on MENTALESE, with average accuracy decreasing by 35.5 p.p. relative to the base model. This decline stems from the fact that SFT encourages the model to restructure reasoning into a symbolic MENTALESE format, typically resulting in a single linear reasoning path. In contrast, DeepSeek R1-style reasoning traces often include "forking tokens" such as *wait*,

Table 1: Model performance across benchmarks. Each block shows Pass@1, average tokens, and compression rate (CR).

| Model | AIME2024 | | | AIME2025 | | | MATH | | | AMC | | | MINERVA | | | OLYMPIAD | | |
|---|---|---|---|---|---|---|---|---|---|---|---|---|---|---|---|---|---|---|
| | P@1 | Tokens | CR | P@1 | Tokens | CR | P@1 | Tokens | CR | P@1 | Tokens | CR | P@1 | Tokens | CR | P@1 | Tokens | CR |
| *Baselines* | | | | | | | | | | | | | | | | | | |
| L1 Max | 0.04 | 390 | 19.18 | 0.03 | 424 | 17.99 | 0.61 | 220 | 11.57 | 0.38 | 310 | 19.79 | 0.17 | 190 | 21.66 | 0.24 | 290 | 23.65 |
| GPT-4o | 0.10 | 923 | 8.11 | 0.06 | 998 | 7.65 | 0.78 | 607 | 4.19 | 0.52 | 1003 | 6.12 | 0.49 | 837 | 4.92 | 0.42 | 956 | 7.17 |
| Claude 3.5 Sonnet | 0.16 | 819 | 9.14 | 0.03 | 764 | 9.99 | 0.75 | 458 | 5.56 | 0.47 | 775 | 7.92 | 0.46 | 694 | 5.93 | 0.36 | 701 | 9.78 |
| Llama3.3 70B | 0.25 | 1114 | 6.72 | 0.03 | 1067 | 7.15 | 0.74 | 516 | 4.93 | 0.47 | 845 | 7.26 | 0.36 | 736 | 5.59 | 0.36 | 780 | 8.79 |
| *Base Models and our Finetuned Variants* | | | | | | | | | | | | | | | | | | |
| DeepSeek R1 1.5b | 0.19 | 7481 | 1.00 | **0.2** | 7631 | 1.00 | 0.77 | 2545 | 1.00 | 0.51 | 6136 | 1.00 | 0.23 | 4115 | 1.00 | 0.35 | 6858 | 1.00 |
| + SLPO (Direct) | **0.26** | 6510 | 1.15 | **0.2** | 5910 | 1.29 | **0.78** | 1017 | 2.50 | **0.60** | 2797 | 2.19 | **0.24** | 1393 | 2.96 | **0.41** | 3031 | 2.26 |
| *MentaSeek* | 0.004 | 184 | 40.65 | 0.01 | 136 | 56.10 | 0.75 | 156 | 16.31 | 0.51 | 168 | 36.52 | 0.12 | 147 | 27.99 | 0.12 | 157 | 43.68 |
| *MentaSeek-GRPO* | 0.2 | 571 | 13.10 | 0.13 | 524 | 14.56 | **0.75** | 301 | 8.45 | **0.51** | 459 | 13.37 | **0.26** | 317 | 12.98 | **0.36** | 439 | 15.68 |
| *MentaSeek-SLPO* | **0.23** | 976 | 7.66 | **0.16** | 785 | 9.72 | 0.61 | 68 | 37.43 | 0.44 | 439 | 13.98 | 0.18 | 86 | 47.85 | 0.32 | 453 | 26.68 |
| Agentica-24k | **0.36** | 7049 | 1.06 | **0.27** | 7270 | 1.05 | **0.86** | 1841 | 1.38 | 0.67 | 4062 | 1.51 | **0.29** | 3466 | 1.19 | **0.47** | 4377 | 1.57 |
| + SLPO (Direct) | 0.33 | 4379 | 1.71 | 0.23 | 3889 | 1.95 | 0.83 | 504 | 5.05 | **0.68** | 1696 | 3.59 | 0.27 | 640 | 6.43 | 0.46 | 1668 | 4.11 |
| *Magentica* | 0.01 | 126 | 59.37 | 0.01 | 126 | 60.56 | 0.40 | 97 | 26.23 | 0.14 | 128 | 47.94 | 0.13 | 119 | 34.58 | 0.12 | 131 | 52.34 |
| *Magentica-GRPO* | 0.19 | 649 | 11.52 | 0.12 | 541 | 14.11 | 0.77 | 257 | 9.91 | 0.53 | 493 | 12.44 | 0.26 | 266 | 15.47 | 0.37 | 453 | 15.13 |
| *Magentica-SLPO* | 0.24 | 1307 | 5.72 | 0.21 | 971 | 7.86 | 0.81 | 424 | 5.99 | 0.6 | 966 | 6.35 | **0.29** | 542 | 7.59 | 0.43 | 889 | 7.71 |

Table 2: Model performance on out-of-domain (OOD) benchmarks GPQA, LSAT, MMLU-PRO and Average. Each block shows Pass@1, average tokens, and compression rate (CR%).

| Model / Training Regime | GPQA | | | LSAT | | | MMLU-1000 | | | Average | | |
|---|---|---|---|---|---|---|---|---|---|---|---|---|
| | P@1 | Tokens | CR% | P@1 | Tokens | CR% | P@1 | Tokens | CR% | P@1 | Tokens | CR% |
| *Baselines* | | | | | | | | | | | | |
| L1 Max | 0.29 | 518 | 14.58 | 0.22 | 554 | 14.79 | 0.45 | 288 | 2.46 | 0.32 | 453 | 10.61 |
| *Base Models and our Finetuned Variants* | | | | | | | | | | | | |
| DeepSeek R1 1.5b | 0.31 | 7551 | 1.00 | 0.24 | 8192 | 1.00 | 0.45 | 710 | 1.00 | 0.33 | 5484 | 1.00 |
| + SLPO (Direct) | **0.33** | 3434 | 2.20 | 0.25 | 5608 | 1.46 | 0.45 | 490 | 1.45 | 0.34 | 3177 | 1.70 |
| *MentaSeek* | 0.25 | 142 | 53.21 | 0.2 | 316 | 25.92 | 0.35 | 67 | 10.60 | 0.27 | 175 | 29.91 |
| *MentaSeek-GRPO* | 0.31 | 366 | 20.64 | 0.23 | 587 | 13.95 | **0.46** | 166 | 4.28 | 0.33 | 373 | 12.96 |
| *MentaSeek-SLPO* | 0.32 | 393 | 19.22 | 0.23 | 851 | 9.63 | 0.43 | 49 | 14.49 | 0.33 | 431 | 14.45 |
| Agentica-24k | 0.36 | 5010 | 1.51 | 0.29 | 7614 | 1.08 | 0.48 | 934 | 0.76 | 0.38 | 4519 | 1.12 |
| + SLPO (Direct) | **0.33** | 2101 | 3.59 | **0.28** | 4077 | 2.01 | **0.48** | 390 | 1.82 | **0.36** | 2189 | 2.47 |
| *Magentica* | 0.26 | 135 | 55.93 | 0.21 | 201 | 40.76 | 0.35 | 62 | 11.45 | 0.27 | 133 | 36.05 |
| *Magentica-GRPO* | 0.30 | 362 | 20.85 | 0.23 | 564 | 14.52 | 0.47 | 157 | 4.52 | 0.33 | 361 | 13.30 |
| *Magentica-SLPO* | **0.33** | 735 | 10.28 | 0.25 | 1200 | 6.83 | **0.48** | 282 | 2.52 | 0.35 | 739 | 6.54 |

*but*, or *so*, which allow the model to self-verify and revise its reasoning mid-generation—boosting accuracy through exploratory pathways. The strict structure imposed by SFT sacrifices these benefits, limiting the model's flexibility and test-time scaling. However, applying RLVR largely reverses this effect: models regain most of the lost accuracy while maintaining significantly shorter reasoning traces—typically just a few hundred tokens longer than their SFT outputs. This highlights the complementary roles of the two stages: SFT enforces symbolic conciseness, while RLVR restores accuracy by reintroducing adaptive reasoning behaviors within that compact framework.

**Training Time Efficiency.** Large-scale reinforcement learning typically demands substantially more computational resources than supervised fine-tuning. In our experiments, we found that applying RLVR directly on the base models required 5–6 days of training on our dataset of 40k samples for 1500 RL steps. This inefficiency arises primarily from the generation of long reasoning chains (often exceeding 8k tokens), which introduces high latency in the vLLM inference engine and becomes the main bottleneck of RLVR training. The cost further increases with larger rollout group sizes. As shown in Figure 3, introducing an intermediate supervised fine-tuning stage on the MENTALESER dataset significantly reduced training cost by 7–10×, while achieving performance close to the base model but with 10× shorter reasoning traces. This demonstrates that aligning models to a more compact reasoning language before RL training not only improves efficiency but also provides a scalable mechanism for reinforcement learning in reasoning tasks.

**Training Collapse Under Direct SLPO.** As shown in Figure 3, applying SLPO directly to AGENTICA-24K resulted in a sudden collapse after approximately 300 training steps. Initially, the response length decreased by nearly half while accuracy improved marginally. However, beyond this point, the average response length rapidly expanded to the maximum generation limit (8k tokens), and the gradient norm curve exhibited NaN values. This instability ultimately caused

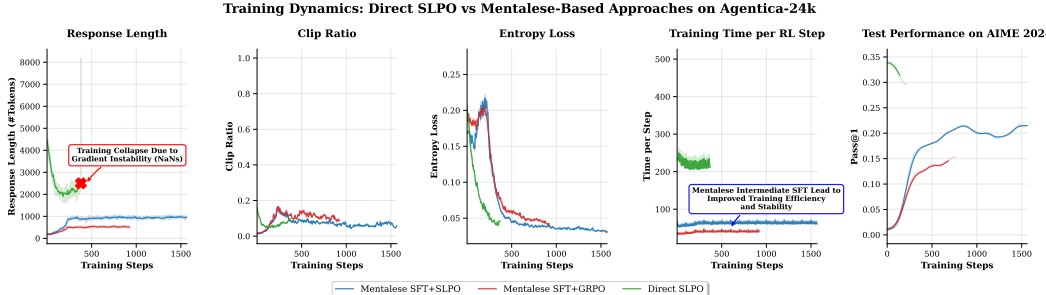

Figure 3: This figure compares direct SLPO on the base model with Intermediate SFT followed by RLHF methods (SLPO/GRPO) on the MENTALESER-40K dataset across five metrics. The *Mentalese* alignment yields greater training stability and efficiency.: (1) Response Length reveals direct SLPO collapses due to gradient instability, while *MentaLMs* stay stable; (2) Clip Ratio indicates more controlled updates in Mentalese methods, driven by reduced response truncation.; (3) Entropy Loss reflects better exploration–exploitation balance; (4) Training Time per RL Step shows higher computational efficiency; (5) Test Performance on AIME 2024 ( 20% Pass@1) confirms *MentaLMs* outperform direct SLPO on the base model. Shaded regions denote min–max ranges across runs. These results highlight the importance of structured intermediate representations (*Mentalese*) for stable, efficient RL in large language models.

the training to collapse, highlighting the difficulty of applying SLPO on raw base models without intermediate alignment. In contrast, introducing an intermediate SFT stage on the MENTALESER dataset maintained stability throughout the entire training process, underscoring the reliability of our proposed two-stage approach.

**Reversion to Verbose Reasoning Under Large Generation Budgets During RLVR.** We observed that when the maximum generation length was set to 4k or 8k tokens, the model tended to drift away from the compact reasoning style learned during the SFT stage and revert to its original base behavior of producing verbose chains. In some cases, this even led to model collapse Figure 3. A likely explanation is that longer reasoning traces, although verbose, occasionally produced correct answers and were therefore rewarded, inadvertently steering the model away from the MENTALESE format. To mitigate this effect, we restricted the maximum generation length to 1k tokens for GRPO-based training and 2k tokens for SLPO-based training. These limits preserved the symbolic reasoning behavior acquired during SFT while still allowing sufficient space for problem-solving.

## 6 CONCLUSION

We introduced a cognitively inspired framework for efficient reasoning that combines *Mentalese*, a compact symbolic reasoning format, with *Shorter Length Preference Optimization* (SLPO), a reinforcement learning strategy that adaptively balances conciseness and correctness. Our model achieves over $10\times$ compression in reasoning traces while maintaining accuracy close to that of verbose large reasoning models, reducing both training and inference costs significantly. By aligning models toward concise and structured reasoning, we provide a pathway for deploying large reasoning capabilities within real-time and resource-constrained environments. Our results suggest that reasoning does not inherently require verbosity, and that carefully designed representations and optimization objectives can yield models that reason more like humans—symbolically, compositionally, and efficiently. This is particularly valuable for agentic systems, where efficient and reliable decision-making is critical, and inference overhead can quickly become a bottleneck.

## 7 ETHICS STATEMENT

We read the ICLR Code of Ethics before the submission. Our paper introduces cognitively inspired compressed reasoning to improve efficiency in large reasoning models (LRMs), reducing computational costs and latency while promoting broader accessibility. At the same time, compressed reasoning may reduce interpretability, and efficiency gains could encourage premature deployment

in high-stakes domains. Our work also inherits common risks of large models, including bias and hallucination, and should therefore be applied responsibly with human oversight.

## 8 REPRODUCIBILITY

We specify the experimental setting in Appendix A.2.2 and attached the training and evaluation codebase along with data as supplementary materials.

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

# A  APPENDIX

You may include other additional sections here.

## A.1  LLM USAGE

We used LLM to help with paper writing for improving grammar and wording, and to help us write code for building plots.

## A.2  METRIC FORMULATION AND HYPERPARAMETERS

### A.2.1  METRICS

- **Pass@1:** Defined as

$$\text{pass@1} = \frac{1}{k} \sum_{i=1}^{k} p_i,$$

(3)

  where $p_i$ denotes whether the $i$-th problem was solved correctly, and $k$ is the total number of test problems. Intuitively, pass@1 measures the fraction of correctly solved problems under single-sample decoding.

- **Token Length:** The average number of tokens produced per response for a given benchmark. We first compute the mean output length across all questions in the dataset, and then report the overall average.

- **Compression Rate (CR):** Defined as

$$\text{CR} = \frac{L_{\text{DeepSeek-R1-1.5B}}}{L_{\text{model}}},$$

  where $L$ denotes the average response length in tokens for a given benchmark. A higher CR indicates greater compression. For example, $\text{CR} = 10$ means that the model generates responses ten times shorter than DEEPSEEK-R1-1.5B.

### A.2.2  HYPERPARAMETERS

Table 3: Key Hyperparameters for Supervised Fine-tuning

| Parameter | Value | Description |
|---|---|---|
| **Data Configuration** | | |
| Cutoff Length | 15000 | Maximum sequence length |
| Validation Split | 0.05 | Fraction of data used for validation |
| **Model & Training** | | |
| Fine-tuning Type | Full | Full parameter fine-tuning |
| Learning Rate | 1e-6 | Optimizer learning rate |
| LR Scheduler | Cosine | Learning rate scheduling strategy |
| Warmup Steps | 20 | Number of warmup steps |
| Train Epochs | 5 | Total number of training epochs |
| **Batch Configuration** | | |
| Per Device Train Batch | 1 | Training batch size per device |
| Gradient Accumulation | 2 | Steps to accumulate gradients |
| **Evaluation & Saving** | | |
| Save Strategy | Steps | Save checkpoints by steps |
| Save Steps | 0.20 | Save every 20% of total steps |
| Eval Steps | 0.05 | Evaluate every 5% of total steps |
| **System Configuration** | | |
| Template | DeepSeek-R1 | Model template/format |
| Precision | BF16 | Mixed precision training |
| Flash Attention | FA2 | Attention optimization |

Table 4: Key Hyperparameters for RLVR Training

| Parameter | Value | Description |
|---|---|---|
| **Data Configuration** | | |
| Train Batch Size | 128 | Batch size for training data |
| Max Prompt Length | 1024 | Maximum length of input prompts |
| Max Response Length | 8192(Direct SLPO), 2048 (SLPO), 1024 (GRPO) | Maximum length of generated responses |
| **Model & Training** | | |
| Learning Rate | 1e-6 | Actor model learning rate |
| PPO Mini Batch Size | 64 | Mini-batch size for PPO updates |
| **KL Divergence Control** | | |
| KL Loss Coefficient | 0.001 | Weight for KL divergence loss |
| KL Loss Type | low_var_kl | Type of KL loss computation |
| **Rollout Configuration** | | |
| Temperature | 0.6 | Sampling temperature for generation |
| Number of Samples | 16 | Samples per prompt during training |
| **Training Schedule** | | |
| Total Epochs | 5 | Number of training epochs |

## A.3 MENTALESER-40K EXAMPLES

### Example 1

```
Instruction:
The operation $\\otimes$ is defined for all nonzero numbers by $a \\otimes b = \\frac{a^{2}}{b}$.
    Determine $[(1 \\otimes 2) \\otimes 3] - [1 \\otimes (2 \\otimes 3)]$.

Mentalese: <think>DEF:(a,b)=a^2/b;CALC1:x=12=1/2;CALC2:y=x3=(1/2)^2/3=1/12;CALC3:z=23=4/3;
    CALC4:w=1z=1/(4/3)=3/4;SUB:y-w=1/12-3/4=-8/12=-2/3; ANS:-2/3</think>\\boxed{-\\frac{2}{3}}
```

### Example 2

```
Instruction:
Doug constructs a square window using $8$ equal-size panes of glass. The ratio of the height to width
    for each pane is $5 : 2$, and the borders around and between the panes are $2$ inches wide. In
    inches, what is the side length of the square window?

Mentalese:
<think>LET:x;PANE_HEIGHT:5x;PANE_WIDTH:2x;ROWS:2;COLS:4;HEIGHT_PANES:2*5x=10x;WIDTH_PANES:4*2x=8x;
    HEIGHT_BORDERS:3*2=6;WIDTH_BORDERS:5*2=10;TOTAL_HEIGHT:10x+6;TOTAL_WIDTH:8x+10;EQ:10x+6=8x+10;
    SOLVE:x=2;WINDOW_SIDE:10*2+6=26;ANS:26</think>\\boxed{26}
```

### Example 3

```
Instruction:
Let $P(x)$ be a polynomial of degree $3n$ such that\n\\begin{align*} P(0) = P(3) = \\dots = P(3n) &=
    2, \\\\ P(1) = P(4) = \\dots = P(3n+1-2) &= 1, \\\\ P(2) = P(5) = \\dots = P(3n+2-2) &= 0.
    \\end{align*}\nAlso, $P(3n+1) = 730$. Determine $n$.

Mentalese:
<think>DEF:P(x),deg=3n;SET:P(3k)=2,k=0..n;SET:P(3k+1)=1,k=0..n-1;SET:P(3k+2)=0,k=0..n-1;COUNT:pts=3n+1;
    EQ:deg+1=pts;SOLVE:n=1;ANS:1</think>\\boxed{1</answer}
```

### Example 4

```
Instruction:
At Euclid Middle School the mathematics teachers are Mrs. Germain, Mr. Newton, and Mrs. Young. There
    are $11$ students in Mrs. Germain's class, $8$ students in Mr. Newton's class, and $9$ students
    in Mrs. Young's class taking the AMC $8$ this year. How many mathematics students at Euclid
    Middle School are taking the contest?

Mentalese:
If $991+993+995+997+999=5000-N$, then $N=$", "output":
    "<think>SUM:s=991+993+995+997+999;CALC:s=4975;EQ:4975=5000-N;SOLVE:N=5000-4975;CALC:N=25;
    ANS:25</think>\\boxed{25</answer}
```

### Example 5

```
Instruction:
Let $f$ be the function defined by $f(x)=ax^2-\\sqrt{2}$ for some positive $a$. If
    $f(f(\\sqrt{2}))=-\\sqrt{2}$ then $a=$

Mentalese:
<think>DEF:f(x)=ax^2-2;EVAL1:f(2)=2a-2;EVAL2:f(f(2))=a(2a-2)^2-2;EXPAND1:(2a-2)^2=4a^2-4a2+2;
    EVAL3:a(4a^2-4a2+2)-2=4a^3-4a22+2a-2;SET:4a^3-4a22+2a-2=-2;ADD:4a^3-4a22+2a=0;
    FACTOR:2a(2a^2-2a2+1)=0;SOLVE:2a^2-2a2+1=0;a=[22(8-8)]/4;a=2/2;
    ANS:2/2</think>\\boxed{\\frac{\\sqrt{2}}{2}}
```

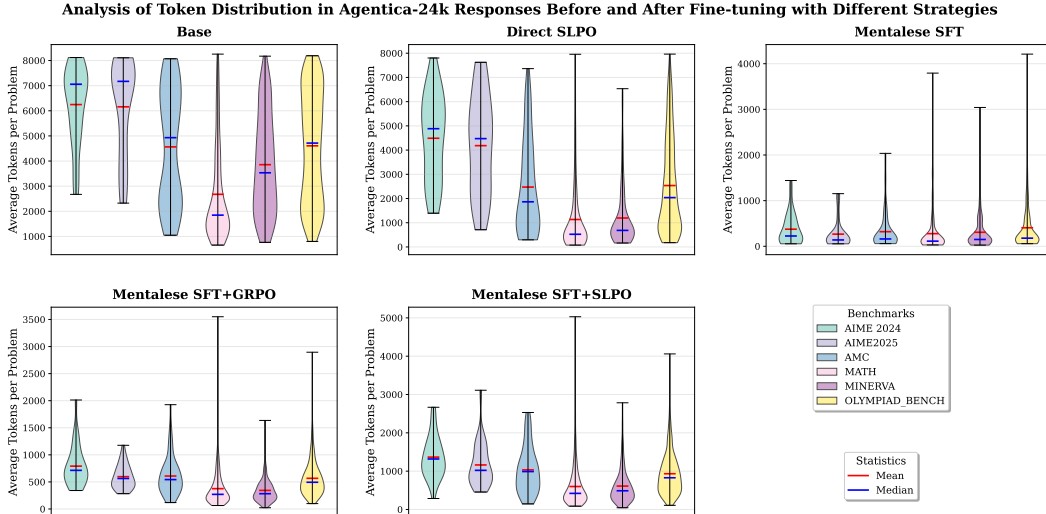

Figure 4: Violin plots of token usage in Agentica-24k responses across six benchmarks before and after fine-tuning. The Base model generates very long responses, while Direct SLPO provides only limited compression. Mentalese-based methods (SFT, SFT+GRPO, SFT+SLPO) achieve 10–20× reduction in response length, approaching an optimal reasoning length that balances efficiency with performance. Although some performance degradation occurs, the Mentalese training pipeline with RLVR methods offers the best trade-off between token efficiency and problem-solving ability.

## A.4 PROMPT STRUCTURE

---

**Direct Prompting Baseline**

**System prompt:**

You are a mathematical reasoning and cognitive modeling assistant. Your job is to generate a detailed step-by-step solution, and then compress it into the given format for cognitive modeling. You are always given the final answer to guide your solution.
Please output the following:

1. **Mentalese CoT** – A symbolic, logic-based chain of thought using compact primitives and logical steps only. Each step should be separated by a semicolon (`;`) with **no natural language**. All core logic steps must be preserved.

**Mentalese Format Rules**

- Use **ONLY** symbolic notation in the form: `OP:params` or `OP:result`
- No natural language
- No redundancy
- Every step must be **necessary and sufficient** for solving the problem
- Example:

  SET:w;EQ:abs(180-5.5*w)=110;CASE1:180-5.5*w=110;SOLVE1:w=70/5.5;
  CASE2:180-5.5*w=-110;SOLVE2:w=290/5.5;CALC1:w1=70/5.5=12+8/11;
  CALC2:w2=290/5.5=52+8/11;DIFF:t=w2-w1=40;ANS:40

Wrap the output in a single JSON object with the following keys:
{ "mentalese_cot"}

**User:**
QUESTION: [Your question here]
ANSWER: [Your answer here]

RESPONSE:

---

## A.5 MODEL RESPONSES

---

**Question (Taken from Math-500)**

What is the smallest positive perfect cube that can be written as the sum of three consecutive integers?

---

**Ground Truth**

27

---

**Magentica**                             **Length: 122 Tokens**

```
<think>\nSET:n;SUM:n+(n+1)+(n+2)=3n+3;EQ:3n+3=k^3;DIV:3n+3=3(n+1);
    SET:k=3m;EQ:3(n+1)=(3m)^3=27m^3;EQ:n+1=9m^3;SOLVE:n=9m^3-1; CHECK:smallest
    m=1;CALC:n=8;SUM:8+9+10=27;ANS:27\n</think>\n\n\\boxed{27}
```

---

**Magentica-SLPO**                          **Length: 351 Tokens**

```
<think>\nGiven: Find the smallest positive perfect cube that can be written as the sum of three
    consecutive integers.\n\nLet me denote the three consecutive integers as n-1, n, n+1. Their sum
    is (n-1) + n + (n+1) = 3n.\n\nSo, we need 3n to be a perfect cube.\n\nLet me denote the perfect
    cube as k^3. So, 3n = k^3 => n = k^3 / 3.\n\nSince n must be an integer, k^3 must be divisible
    by 3. Therefore, k must be divisible by 3. Let me set k = 3m, where m is an integer.\n\nThen,
    k^3 = (3m)^3 = 27m^3.\n\nSo, 3n = 27m^3 => n = 9m^3.\n\nTherefore, the sum of three consecutive
    integers is 3n = 27m^3, which is a perfect cube.\n\nNow, we need the smallest positive perfect
    cube. Since m is a positive integer, the smallest m is 1.\n\nTherefore, the smallest perfect
    cube is 27*1^3 = 27.\n\nLet me verify: Find three consecutive integers that sum to 27.\n\nn =
    9m^3 = 9*1 = 9.\n\nSo, the integers are 8, 9, 10. Their sum is 8+9+10=27.\n\nYes, that
    works.\n\nTherefore, the smallest positive perfect cube is 27.\n</think>\n\n\\boxed{27}
```

---

**Magentica-GRPO**                          **Length: 236 Tokens**

```
<think>Let's denote the three consecutive integers as n-1, n, n+1. Their sum is (n-1)+n+(n+1)=3n. We
    need 3n to be a perfect cube. So, n must be a multiple of 3. Let n=3k. Then 3n=9k. We need 9k
    to be a perfect cube. Let's write 9k=m^3. Then k=m^3/9. For k to be an integer, m^3 must be
    divisible by 9, so m must be divisible by 3. Let m=3t. Then m^3=27t^3. So k=27t^3/9=3t^3.
    Therefore, n=3k=9t^3. Then 3n=27t^3=(3t^3)^3? Wait, no. 27t^3 is (3t)^3. Yes, 27t^3=(3t)^3. So
    the smallest positive perfect cube is when t=1, so 27.\n</think>\n\n\\boxed{27}
```

---

**Agentic-24k-SLPO**                        **Length: 526 Tokens**

```
<think>\nOkay, so I need to find the smallest positive perfect cube that can be written as the sum of
    three consecutive integers. Hmm, let's break this down step by step.\n\nFirst, let's denote the
    three consecutive integers. Let's say the integers are n-1, n, and n+1. So their sum would be
    (n-1) + n + (n+1). Simplifying that, it's n-1 + n + n+1 = 3n. So the sum of three consecutive
    integers is 3n.\n\nNow, we need this sum to be a perfect cube. So, 3n must be a perfect cube.
    Let's denote the perfect cube as k, where k is a positive integer. Therefore, 3n = k, which
    means n = k / 3.\n\nSince n has to be an integer (because we're dealing with integers), k must
    be divisible by 3. That implies that k itself must be divisible by 3, because if 3 divides k,
    then 3 must divide k. So let's let k = 3m, where m is a positive integer.\n\nSubstituting back,
    we have k = (3m) = 27m. Therefore, n = 27m / 3 = 9m.\n\nSo, the sum of three consecutive
    integers is 3n = 3 * 9m = 27m. Wait, that seems a bit off. Let me correct that.\n\nIf n = 9m,
    then the sum is 3n = 27m. So, the sum is 27m, which is a perfect cube since 27 is 3 and m is a
    cube. Therefore, the sum is (3m), which is indeed a perfect cube.\n\nNow, we need the smallest
    positive perfect cube. So we need the smallest m such that 27m is positive. Since m is a
    positive integer, the smallest m is 1.\n\nTherefore, the smallest positive perfect cube is 27 *
    1 = 27.\n\nLet me verify this. If m=1, then n=9*1=9. So the three consecutive integers are
    8,9,10. Their sum is 8+9+10=27, which is 3. That checks out.\n</think>\n\nThe smallest positive
    perfect cube that can be written as the sum of three consecutive integers is \\boxed{27}.
```

---

**Agentica-24k**  **Length: 1512 Tokens**

<think>\nOkay, so I need to find the smallest positive perfect cube that can be written as the sum of three consecutive integers. Hmm, let's break this down step by step.\n\nFirst, let's understand what it means for a number to be a perfect cube. A perfect cube is an integer that can be written as n, where n is an integer. The smallest positive perfect cubes are 1=1, 2=8, 3=27, and so on.\n\nNow, the problem states that this perfect cube must be expressible as the sum of three consecutive integers. Let's denote these three consecutive integers as n-1, n, and n+1. Wait, actually, maybe it's better to represent them as x, x+1, x+2, where x is an integer. That might be more straightforward.\n\nSo, the sum of three consecutive integers can be written as x + (x+1) + (x+2). Let's compute this:\n\nx + (x+1) + (x+2) = x + x + 1 + x + 2 = 3x + 3 = 3(x + 1).\n\nSo, the sum of three consecutive integers is equal to 3 times (x + 1). That means that the sum must be a multiple of 3.\n\nNow, since we're looking for a perfect cube, let's denote the perfect cube as k, where k is a positive integer. So, we have:\n\nk = 3(x + 1).\n\nTherefore, k must be divisible by 3. Which implies that k itself must be divisible by 3, because if a prime number divides a cube, it must divide the base. Since 3 is prime, k must be a multiple of 3. Let's denote k as 3m, where m is a positive integer.\n\nSubstituting back, we get:\n\n(3m) = 3(x + 1)\n\nCompute the left side:\n\n27m = 3(x + 1)\n\nDivide both sides by 3:\n\n9m = x + 1\n\nTherefore, x + 1 = 9m, which implies x = 9m - 1.\n\nNow, since x is an integer, and we're looking for positive integers, let's see what m can be.\n\nWe need k to be positive, so k must be at least 1. But since k is a multiple of 3, the smallest possible k is 3.\n\nLet's compute the smallest possible k, which would be when m=1, because m=1 gives k=3.\n\nSo, k=3, then k=27.\n\nNow, let's check if 27 can be expressed as the sum of three consecutive integers.\n\nFrom earlier, we know that the sum is 3(x + 1). So, 27 = 3(x + 1) => x + 1 = 9 => x = 8.\n\nTherefore, the three consecutive integers are x=8, x+1=9, x+2=10. Let's check their sum: 8+9+10=27. Yes, that works.\n\nBut wait, let's confirm if 27 is indeed the smallest such perfect cube. Let's check smaller k values that are multiples of 3.\n\nThe next smaller k would be k=0, but that would give k=0, which isn't positive. Then k=3 is the next one, which gives k=27. So, 27 is indeed the smallest positive perfect cube that can be written as the sum of three consecutive integers.\n\nBut just to be thorough, let's check if there's any smaller k, perhaps non-multiples of 3? Wait, earlier reasoning showed that k must be a multiple of 3 because k must be divisible by 3. Therefore, the smallest k is 3, leading to k=27. So, yes, 27 is the answer.\n\nWait, hold on, let me think again. The problem says \"the smallest positive perfect cube that can be written as the sum of three consecutive integers.\" So, perhaps I should check if there's a smaller cube than 27 that can be expressed as such.\n\nLet's consider the cubes:\n\nn1=1: Can 1 be written as the sum of three consecutive integers? Let's see: Let x + (x+1) + (x+2) = 3x + 3 =1. Then 3x= -2 => x= -2/3. Not an integer, so no.\n\nn2=8: Let's see if 8 can be written as the sum of three consecutive integers. Let's set up the equation: 3x + 3 =8 => 3x=5 => x=5/3. Not an integer. So, no.\n\n3=27: As before, x=8, which is an integer, so yes.\n\nSo, indeed, 27 is the smallest positive perfect cube that can be written as the sum of three consecutive integers.\n\nTherefore, the answer is 27.\n\n**Final Answer**\nThe smallest positive perfect cube is \\boxed{27}.\n</think>\n\nTo find the smallest positive perfect cube that can be written as the sum of three consecutive integers, we start by noting that the sum of three consecutive integers can be expressed as \\( x + (x+1) + (x+2) = 3x + 3 = 3(x + 1) \\).\n\nThis sum must be a perfect cube, denoted as \\( k^3 \\). Therefore, we have:\n\\[ k^3 = 3(x + 1) \\]\n\nSince \\( k^3 \\) must be divisible by 3, \\( k \\) itself must be a multiple of 3. Let \\( k = 3m \\) where \\( m \\) is a positive integer. Substituting this, we get:\n\\[ (3m)^3 = 3(x + 1) \\]\n\\[ 27m^3 = 3(x + 1) \\]\n\\[ 9m^3 = x + 1 \\]\n\\[ x = 9m^3 - 1 \\]\n\nWe need the smallest \\( k \\) such that \\( k^3 \\) is positive. The smallest \\( k \\) is 3, leading to:\n\\[ k^3 = 3^3 = 27 \\]\n\nWe verify that 27 can be expressed as the sum of three consecutive integers:\n\\[ 3(x + 1) = 27 \\]\n\\[ x + 1 = 9 \\]\n\\[ x = 8 \\]\n\nThus, the three consecutive integers are 8, 9, and 10, whose sum is 27.\n\nTherefore, the smallest positive perfect cube is \\(\\boxed{27}\\).

