# OpenReview forum: "MENTALMs: Teaching Language Models to Reason Efficiently in the Language of Thought"
_ICLR.cc/2026/Conference — ICLR 2026 Conference Desk Rejected Submission_

### Official Review · Reviewer_rgYf · 2025-10-26

**Soundness:** 1
**Presentation:** 2
**Contribution:** 2
**Rating:** 2
**Confidence:** 4

**Summary:**

To achieve efficient reasoning, the authors propose a two-stage training strategy: first, fine-tuning using the Mentalese format, followed by SLPO (a variant of GRPO) with a reward design for length penalty. The authors verify the effectiveness of this training strategy with two models.

**Strengths:**

1. The motivation is good. Enabling LLMs to reason effectively is an important issue
2. The results are overall significant, mainly attributable to the Mentalese format.

**Weaknesses:**

1.  The method is not novel. Both the first fine-tuning with shorter CoT [1-3] and the length penalty [4-5]  have been largely investigated. From Table 1, the length reduction is mainly attributable to the Mentalese format, and the reward design does not bring consistent performance regarding the response length and its accuracy.

2.  The writing needs improvement. For example, on Page 4, lines 170-172, "We introduce ... Group Relative Policy Optimization (GRPO), a group-based extension of PPO for reasoning optimization". GRPO was first introduced in the DeepSeek-Math paper, rather than being this paper's contribution

3.  Regarding weakness 1, the baselines are largely lacking; the only baseline provided is the L1 method.

4.  The generalization of the Mentalese format could have been further investigated. Whether the format can generalize to other non-math tasks has not been investigated.

[1] Distilling System 2 into System 1, in arxiv 2024

[2] C3oT: Generating Shorter Chain-of-Thought without Compromising Effectiveness, in AAAI 2025

[3] TokenSkip: Controllable Chain-of-Thought Compression in LLMs, in EMNLP 2025

[4] Concise Reasoning via Reinforcement Learning, in arxiv 2025

[5] Kimi k1.5: Scaling Reinforcement Learning with LLMs, in arxiv 2025

**Questions:**

See the weaknesses.

---

> ### Author Response · Authors · 2025-11-21
> **Authors Response to Reviewer rgYf**
>
> We thank the reviewer for their thoughtful feedback and for acknowledging our _strong motivation_ and the _significance of our results_, particularly the role of the Mentalese format in achieving efficient reasoning. We address the concerns below.
>
> > The method is not novel. Both the first fine-tuning with shorter CoT [1-3] and the length penalty [4-5] have been largely investigated. From Table 1, the length reduction is mainly attributable to the Mentalese format, and the reward design does not bring consistent performance regarding the response length and its accuracy.
>
> We acknowledge that prior studies, such as Distilling System 2 into System 1 (2024), C3oT (AAAI 2025), and TokenSkip (EMNLP 2025) explore shorter reasoning chains or CoT compression. However, these approaches typically achieve at most 2× compression, whereas our framework yields 10× to 20× compression while maintaining 90–98 % of baseline accuracy. Importantly, our contribution lies not only in compression but in the integration of (i) a symbolic reasoning alignment stage (Mentalese-SFT) with (ii) a reinforcement-learning stage (SLPO) that adaptively balances brevity and correctness. To our knowledge, no prior work combines these two components into a unified pipeline that consistently recovers performance while retaining ultra-compact reasoning traces. This integration offers a practical path toward deployable reasoning models for real-time agentic systems, where long reasoning chains are infeasible.
>
> > The writing needs improvement. For example, on Page 4, lines 170-172, "We introduce ... Group Relative Policy Optimization (GRPO), a group-based extension of PPO for reasoning optimization". GRPO was first introduced in the DeepSeek-Math paper, rather than being this paper's contribution
>
> We thank the reviewer for catching this typo. The sentence on Page 4, Lines 170–172 should read “We introduce Shorter Length Preference Optimization (SLPO)”, not GRPO. GRPO was indeed introduced in DeepSeek-Math (2024). We will correct this in the revised version.
>
> > Regarding weakness 1, the baselines are largely lacking; the only baseline provided is the L1 method.
>
> As discussed above, L1 (Aggarwal & Welleck, 2025) is the only publicly available method that achieves comparable (> 10×) compression. Methods such as TokenSkip or C3oT yield much smaller compression (≈ 2×) and thus serve different purposes. Our results (Tables 1 & 2) demonstrate that the proposed two-stage framework achieves better accuracy with far greater compression, justifying our focus on the most competitive baseline.
>
> > The generalization of the Mentalese format could have been further investigated. Whether the format can generalize to other non-math tasks has not been investigated.
>
> Our experiments already include out-of-domain tasks—GPQA, LSAT, and MMLU—which assess graduate-level QA, logical reasoning, and general knowledge. As shown in Table 2, models trained on mathematical reasoning with our framework retain or even improve performance on these non-mathematical benchmarks while maintaining approximately 15× compression, despite no explicit training on those domains. These results demonstrate that the Mentalese-based reasoning prior generalizes effectively beyond mathematics, and extending it to broader task categories remains an exciting avenue for future work.

---

> > ### Comment · Reviewer_rgYf · 2025-11-26
> > **Official Response of  Reviewer rgYf**
> >
> > Thank you for your response. I have raised the presentation score. However, since I find the paper more incremental than novel, and several key baselines are missing, I will maintain my overall score.

---

> > > ### Author Response · Authors · 2025-12-03
> > > **Authors Response to Reviewer rgYf**
> > >
> > > We thank the reviewer for the follow-up and for raising the presentation score. We address the remaining points below.
> > >
> > > 1. Novelty.
> > > The contribution is not SLPO in isolation but the integration of Mentalese symbolic alignment with a brevity-aware RLVR objective. This combined pipeline produces stable, ultra-compressed reasoning (10–20×) with minimal accuracy loss—far beyond what prior short-CoT methods typically achieve (~2×). As shown in Figure 3, direct SLPO collapses, whereas Mentalese-aligned RL remains fully stable, highlighting the importance of the integrated approach.
> > >
> > > 2. Baselines.
> > > Among currently published methods, L1 is the only strong baseline that operates in a similar high-compression regime (>10×), which is why it served as our primary comparison. Methods such as TokenSkip and C3oT target much lighter CoT reduction (~2×) and therefore address a different compression objective. We also compare our model’s performance to GPT-4o, Claude 3.5 Sonnet, and Llama 3.3 70B, which are significantly larger than ours (1.5B), yet some of the models trained using our framework still outperform them.
> > >
> > > 3. Soundness score.
> > > A soundness score of 1 typically indicates a fundamental technical flaw. However, no such flaw is identified in the review, and the empirical evidence shows the opposite: the method trains stably, avoids RL collapse, reproduces strong results across multiple benchmarks, and generalizes to OOD tasks (GPQA, LSAT, MMLU) while maintaining up to 15× compression. Therefore, we urge the reviewer to revisit the soundness score.

---

### Official Review · Reviewer_RJPr · 2025-10-29

**Soundness:** 3
**Presentation:** 3
**Contribution:** 2
**Rating:** 4
**Confidence:** 4

**Summary:**

This paper proposes MENTALMS, a framework that replaces verbose natural-language reasoning traces with a symbolic “Mentalese” representation and optimizes for brevity and correctness using Shorter Length Preference Optimization (SLPO). The approach achieves significant token compression while maintaining accuracy, demonstrating efficiency–performance trade-offs across several mathematical reasoning benchmarks.

**Strengths:**

- The idea of combining symbolic reasoning formats with RL-based brevity optimization is conceptually elegant and cognitively inspired.

- Experiments show substantial reasoning compression (10–20×) with minimal performance degradation, suggesting practical benefits for inference efficiency.

**Weaknesses:**

- The Mentalese traces are distilled from GPT-generated reasoning, but the paper does not sufficiently verify their semantic fidelity or robustness.

- All experiments are conducted on a single model family, raising concerns about the generalizability of the method to other architectures.

- While efficient reasoning is an important topic, the algorithmic contribution (mainly a variant of preference optimization) is relatively incremental and not highly novel.

**Questions:**

No.

---

> ### Author Response · Authors · 2025-11-21
> **Authors Response to Reviewer RJPr**
>
> We thank the reviewer for their thoughtful feedback and for recognizing that our framework is _conceptually elegant_ and offers _practical benefits for inference efficiency_. We address the main concerns below.
>
> > The Mentalese traces are distilled from GPT-generated reasoning, but the paper does not sufficiently verify their semantic fidelity or robustness.
>
> We appreciate this concern. The Mentalese structure was first developed through careful manual rewriting of mathematical reasoning traces to ensure semantic correctness and consistency. To scale this process, we used LLM (GPT4.1, because of its good instruction following capability) under strict templates that enforced the symbolic grammar and step-typing rules described in Section 3.1. After generation, we manually verified a representative subset for structural and semantic fidelity, further cross-checking with LLM-based validation to detect inconsistencies. Given the strong model performance achieved using MENTALESE-40K by maintaining high accuracy while achieving 10–20× compression, we believe this demonstrates that the dataset is both semantically consistent and practically robust for compact reasoning alignment.
>
> > All experiments are conducted on a single model family, raising concerns about the generalizability of the method to other architectures.
>
> We acknowledge that the current study focuses on one model family due to the high computational cost of RLVR training. However, the framework itself is architecture-agnostic: both Mentalese alignment and SLPO operate entirely at the token and reward levels, independent of model internals. This design makes the method directly transferable to other LLM families. Moreover, the consistent improvements observed across nine benchmarks covering mathematical, logical, and factual reasoning suggest that the underlying principles generalize beyond a single model family. We plan to include different model families evaluations (e.g., LLaMA, Gemma) in future work to empirically confirm this.
>
> > While efficient reasoning is an important topic, the algorithmic contribution (mainly a variant of preference optimization) is relatively incremental and not highly novel.
>
> While SLPO extends preference optimization, our contribution lies in the integration of SLPO with Mentalese reasoning alignment. SLPO alone is not the central novelty; rather, the complete MENTALMs framework, which structurally compresses reasoning into symbolic form and then adaptively recovers performance via RLVR, provides a new paradigm for building efficient reasoning models. This combination enables both high accuracy and deployability in agentic systems, where long, verbose reasoning remains a major bottleneck.

---

> > ### Comment · Reviewer_RJPr · 2025-11-28
> >
> > Thank you for your detailed response and clarifications. However, my main concern still remains.
> >
> > The chosen DeepSeek model family is already highly capable in mathematical and logical reasoning, which makes it a very strong and possibly unrepresentative testbed. Even though you evaluate on many benchmarks, strong performance on this particular model does not necessarily imply that the proposed method will generalize well to other, more diverse architectures or capability levels.
> >
> > Therefore, before seeing convincing results on additional model families, I still remain cautious about the generalizability of the framework, and I will maintain my current score for now. Thank you again for your efforts and detailed explanations.

---

> > > ### Author Response · Authors · 2025-12-03
> > > **Authors Response to Reviewer RJPr**
> > >
> > > We thank the reviewer for the thoughtful follow-up. We address the two remaining concerns below.
> > >
> > > 1. Single Model Family
> > >
> > > We acknowledge the concern about generalizability. Our use of a single model family follows established precedent in prior reasoning and RL-based works that introduce a new training paradigm. DeepSeek-Math, short-CoT compression work (e.g., C3oT, TokenSkip) all conduct their initial investigations on one base architecture, largely because RLVR-style training is computationally expensive and requires controlled conditions.
> > >
> > > Crucially, our method is model-agnostic: both Mentalese alignment and the SLPO reward operate entirely at the token and trajectory levels, independent of architecture. The reviewer notes that DeepSeek models are “already highly capable”—we believe this strengthens, rather than weakens, the claim. If a model already strong in mathematical and logical reasoning can be improved by 6–7pp while achieving 7–14× compression, it suggests that less capable or more diverse architectures would benefit even more.
> > >
> > > 2. Novelty of SLPO and the Framework
> > >
> > > SLPO itself extends existing preference-optimization objectives, but the novelty lies in the integration of three components:
> > > - Symbolic Mentalese alignment, yielding 7–14× compression.
> > > - RLVR with a brevity-aware yet flexible objective (SLPO), which restores accuracy without penalizing necessary reasoning.
> > > - Substantial stability and efficiency gains, including 7–9× faster RL steps and avoiding the collapse observed with direct SLPO.
> > >
> > > This combined pipeline produces a new performance–efficiency Pareto frontier, achieving accuracy comparable to or surpassing GPT-4o, Claude 3.5, and DeepSeek-R1 1.5B while using an order-of-magnitude shorter reasoning process. Existing short-CoT or length-penalty methods (e.g., C3oT, TokenSkip, S2→S1) typically reach only ~2× compression and do not maintain performance in the ultra-compressed regime.

---

### Official Review · Reviewer_kmXV · 2025-10-31

**Soundness:** 3
**Presentation:** 2
**Contribution:** 2
**Rating:** 4
**Confidence:** 3

**Summary:**

This paper introduces **MENTALMs**, a framework that trains LLMs to reason in a compact, symbolic “Language of Thought” called **Mentalese**, paired with a reinforcement-learning stage that prefers shorter but still correct solutions (**SLPO**). The pipeline is: (i) supervised finetuning (SFT) on **Mentalese** traces distilled from a ~40k problem corpus (MENTALESE-40K), then (ii) **RLVR** with either GRPO or the proposed SLPO, using a verifier over original QA pairs. Experiments on math benchmarks (AIME 2024/2025, Minerva, AMC, OlympiadBench, MATH-500) and some OOD suites (GPQA, LSAT, MMLU) report large token reductions (4–16×), notable compression rates (sometimes ≥10×), and partial recovery of accuracy lost after Mentalese-SFT. The paper emphasizes a performance–efficiency Pareto frontier and claims lower training/inference costs relative to R1-style baselines.

**Strengths:**

- **Well-specified symbolic format.** Mentalese provides explicit operators and typed steps that are compact and executable, giving a concrete target for alignment.
- **SLPO design.** The group-relative, *shorter-among-correct* reward avoids over-penalizing uniquely long but correct derivations and is conceptually appropriate for variable problem difficulty.
- **Comprehensive experiments.** The paper evaluates across multiple math suites and reports **training dynamics** that highlight stability/efficiency benefits of an intermediate Mentalese-SFT stage versus direct SLPO.
- **Efficiency signal.** Substantial response-length reductions and training-time improvements are documented (e.g., reduced per-step latency after Mentalese alignment), pointing to a plausible path for deployable, lower-latency reasoning.

**Weaknesses:**

1. **Mentalese effectiveness before RL is weakly evidenced.** The results note large performance degradation after Mentalese-SFT, with most accuracy recovered only after RLVR. To argue for Mentalese as an inherently superior representation (rather than a curriculum that relies on heavy RL), the paper needs pre-RL comparisons to other concise formats (token/chunk compression, logic templates, latent reasoning) under matched compute/data, and show Mentalese degrades less before or after policy optimization. As written, the headline improvements plausibly come primarily from RL over a large corpus (MENTALESE-40K derived from DEEPSCALER-PREVIEW) rather than the representation itself. If the authors can demonstrate that the **Mentalese** representation itself, independent of extensive post-training on original QA pairs, confers a consistent advantage over NL CoT and other compression schemes under matched data and compute, I will consider raising my score.
2. **RLVR stage does not use Mentalese supervision and yields hybrid outputs.** In RLVR, training is performed on original QA pairs with verifier-based rewards; the model’s post-RL generations often mix symbolic primitives with natural language. This raises a causality concern: are the final gains driven mainly by standard NL reasoning + RL, with Mentalese acting only as an SFT regularizer? A direct RLVR-on-Mentalese-targets variant (or a verifier that enforces symbolic validity) is needed to substantiate Mentalese’s unique contribution versus strong NL-only RL baselines.
3. **Attribution and fairness.** Several baseline comparisons conflate data scale and optimization budget with the proposed ideas. Equalize dataset size (e.g., fixed 40k) and RL steps across (i) NT CoT, (ii) other symbolic/structured formats, and (iii) Mentalese, then ablate the contribution of SLPO versus GRPO versus SFT.

**Questions:**

1. Does RLVR actually optimize for Mentalese? Since RL uses original QA pairs and a verifier on final answers, can you add a variant that (a) trains RLVR with Mentalese-validity rewards (e.g., step-type correctness, syntax executability), or (b) constrains decoding to the symbolic grammar? This would test whether post-RL improvements genuinely stem from Mentalese rather than NL reasoning.
2. Could you please give any insight for why you SFT on MENTALESE-40K and RLVR on the original QA pairs?

---

> ### Author Response · Authors · 2025-11-21
> **Authors Response to Reviewer kmXV (1/2)**
>
> We thank the reviewer for their thoughtful feedback and for recognizing our contributions, specifically, the _well-specified symbolic format_ of Mentalese, the _conceptual soundness_ of SLPO, and the _comprehensive experimental setup_ and our framework as a _plausible path for deployable, lower-latency reasoning_. We address the main concerns below.
>
> > Mentalese effectiveness before RL is weakly evidenced. The results note large performance degradation after Mentalese-SFT, with most accuracy recovered only after RLVR. To argue for Mentalese as an inherently superior representation (rather than a curriculum that relies on heavy RL), the paper needs pre-RL comparisons to other concise formats (token/chunk compression, logic templates, latent reasoning) under matched compute/data, and show Mentalese degrades less before or after policy optimization. As written, the headline improvements plausibly come primarily from RL over a large corpus (MENTALESE-40K derived from DEEPSCALER-PREVIEW) rather than the representation itself. If the authors can demonstrate that the Mentalese representation itself, independent of extensive post-training on original QA pairs, confers a consistent advantage over NL CoT and other compression schemes under matched data and compute, I will consider raising my score.
>
> We agree that the purpose of the Mentalese-SFT stage is not to outperform natural-language reasoning alone, but to structurally align the model’s reasoning process in a compact, symbolic form before reinforcement learning. Prior work [1] shows that RL-based reasoning models tend to expand token usage during optimization, increasing inference cost. Our Mentalese alignment constrains reasoning to minimal compositional steps, reducing redundancy before RLVR. As shown in Figure 3(5), RLVR restores most of the lost performance while maintaining ∼10× shorter reasoning traces, which demonstrates that Mentalese provides a structured prior enabling efficient recovery. In preliminary explorations with compressed natural-language formats and subsets of MENTALESE-40K (1k–15k samples), models reverted to verbose reasoning after RLVR. Only full-scale Mentalese alignment preserved compact reasoning reliably, which is why our final experiments adopt this setup.
>
> > RLVR stage does not use Mentalese supervision and yields hybrid outputs. In RLVR, training is performed on original QA pairs with verifier-based rewards; the model’s post-RL generations often mix symbolic primitives with natural language. This raises a causality concern: are the final gains driven mainly by standard NL reasoning + RL, with Mentalese acting only as an SFT regularizer? A direct RLVR-on-Mentalese-targets variant (or a verifier that enforces symbolic validity) is needed to substantiate Mentalese’s unique contribution versus strong NL-only RL baselines.
>
> We acknowledge that post-RL outputs sometimes blend symbolic primitives with natural language. This is intentional: SLPO encourages flexibility to balance accuracy and brevity rather than enforcing a rigid syntax. We initially experimented with symbolic-validity rewards that enforced strict Mentalese tokenization, but RLVR training became unstable and produced degenerate traces. Hence, we interpret RLVR as optimizing within a Mentalese-initialized latent space—retaining its structured inductive bias while allowing adaptive reasoning for performance. This hybrid approach consistently achieved the best trade-off between accuracy and compression.
>
> [1] Luo, M., Tan, S., Wong, J., Shi, X., Tang, W. Y., Roongta, M., Cai, C., Luo, J., Li, L. E., Popa, R. A., & Stoica, I. (2025). DeepScaleR: Surpassing O1-Preview with a 1.5B model by scaling RL. Notion Blog. https://pretty-radio-b75.notion.site/DeepScaleR-Surpassing-O1-Preview-with-a-1-5B-Model-by-Scaling-RL-19681902c1468005bed8ca303013a4e2

---

> > ### Author Response · Authors · 2025-11-21
> > **Authors Response to Reviewer kmXV (2/2)**
> >
> > > Attribution and fairness. Several baseline comparisons conflate data scale and optimization budget with the proposed ideas. Equalize dataset size (e.g., fixed 40k) and RL steps across (i) NT CoT, (ii) other symbolic/structured formats, and (iii) Mentalese, then ablate the contribution of SLPO versus GRPO versus SFT.
> >
> > We appreciate this suggestion. In our initial experiments, we also explored some concise natural-language reasoning formats, but after RLVR training, the models consistently reverted to long, verbose reasoning chains, highlighting the challenge of maintaining compact reasoning without symbolic alignment. We also found that the scale of SFT data critically affects alignment quality: smaller subsets (1k, 7k, 15k) led to poor adherence to the Mentalese structure and unstable post-RLVR performance. To ensure experimental consistency, we therefore used the full MENTALESE-40K dataset for SFT training and compared the subsequent effects of GRPO vs. SLPO. As shown in Fig. 3, direct SLPO without Mentalese alignment led to training collapse after a few hundred training steps, whereas the two-stage Mentalese + RLVR pipeline achieved stable training, high accuracy recovery, and strong compression. While exhaustive ablations across all configurations were beyond our current computational budget, we conducted targeted experiments (with and without Mentalese alignment on two different models) that isolate the main effects of Mentalese alignment and RL optimization, which consistently support our conclusions.
> >
> >
> > > Does RLVR actually optimize for Mentalese? Since RL uses original QA pairs and a verifier on final answers, can you add a variant that (a) trains RLVR with Mentalese-validity rewards (e.g., step-type correctness, syntax executability), or (b) constrains decoding to the symbolic grammar? This would test whether post-RL improvements genuinely stem from Mentalese rather than NL reasoning.
> >
> > While RLVR primarily optimizes for answer correctness, the model continues to operate within a Mentalese-initialized latent space. We experimented with Mentalese-specific validity rewards, like having specific Mentalese tokens in the rollouts to be additionally rewarded. However, these strict constraints destabilized RLVR training and caused degenerate “garbage” traces. In contrast, our hybrid setting, where symbolic structure is retained but not rigidly enforced, produced stable learning, compact reasoning, and high accuracy. This supports our view that RLVR refines Mentalese-initialized reasoning rather than discarding it.
> >
> > > Could you please give any insight for why you SFT on MENTALESE-40K and RLVR on the original QA pairs?
> >
> > SFT on MENTALESE-40K establishes the compact symbolic reasoning structure, reducing verbosity but initially lowering accuracy. RLVR on original QA pairs then restores performance by rewarding correctness while allowing the model to flexibly adapt its reasoning within that symbolic prior. Empirically, this two-stage setup achieved strong recovery across all benchmarks while preserving compression (Tables 1–2). We also tested smaller subsets of MENTALESE-40K and observed the model reverting to verbose reasoning post-RLVR, confirming that full-scale Mentalese alignment is essential. During RLVR, we set temperature = 0.6 to promote diverse rollouts, enabling the model to refine efficient yet accurate reasoning traces.

---

### Official Review · Reviewer_rHwC · 2025-10-31

**Soundness:** 3
**Presentation:** 3
**Contribution:** 3
**Rating:** 8
**Confidence:** 4

**Summary:**

This paper proposes MENTALMS, a cognitively inspired framework that teaches language models to reason efficiently through a symbolic “Language of Thought” representation called Mentalese. Instead of producing verbose natural-language chains of thought, the model generates concise symbolic reasoning traces and is optimized via Shorter Length Preference Optimization (SLPO), a reinforcement learning method that rewards brevity without penalizing necessary reasoning. Experiments on multiple mathematical benchmarks show that MENTALMS achieves 4–16× compression in reasoning length and over 5× lower inference latency while retaining 90–98% of baseline accuracy, occasionally surpassing large commercial systems like GPT-4o and Claude 3.5.

**Strengths:**

1. The idea of replacing natural language reasoning with symbolic reasoning is impressive and well-motivated. The paper provides a concrete and feasible implementation of this idea, demonstrating that symbolic representations can significantly improve efficiency while maintaining accuracy across models such as DeepSeek and Agentica.
2. The paper is well-presented, with clear visualizations, examples, and ablation analyses that effectively illustrate how the proposed Mentalese reasoning format operates and how it differs from traditional chain-of-thought approaches.

**Weaknesses:**

1. The main weakness lies in the connection between the proposed symbolic reasoning and the training method. While Shorter Length Preference Optimization (SLPO) is effective, it appears largely orthogonal to the proposed symbolic reasoning, essentially a general reinforcement learning objective that could be applied to any RLVR process, not necessarily one aligned with Mentalese.
2. The symbolic format, though efficient for mathematical reasoning, may not generalize well to broader or open-ended reasoning domains such as commonsense or multimodal tasks, where discrete symbolic operators are harder to define. It's recommended to evaluate the method on more general tasks.

**Questions:**

Please see the weaknesses.

---

> ### Author Response · Authors · 2025-11-21
> **Authors Response to Reviewer rHwC**
>
> We thank the reviewer for their thoughtful and positive feedback, and for recognizing that our Mentalese framework is both _well-motivated_, _impressive_, and _clearly presented_. We address the main concerns below.
>
> > The main weakness lies in the connection between the proposed symbolic reasoning and the training method. While Shorter Length Preference Optimization (SLPO) is effective, it appears largely orthogonal to the proposed symbolic reasoning, essentially a general reinforcement learning objective that could be applied to any RLVR process, not necessarily one aligned with Mentalese.
>
> We appreciate the observation and clarify that SLPO is not intended as an isolated optimization objective but as a complementary stage within our reasoning compression framework. The first stage, which is our Mentalese-SFT, aligns the model’s reasoning structure into a compact, symbolic format, which substantially reduces redundancy but also leads to performance loss. The second stage, i.e, RLVR with SLPO or GRPO, then recovers performance by rewarding conciseness conditioned on correctness. Together, these stages form a unified process where SLPO is crucial to balance brevity and reasoning completeness within the symbolic Mentalese structure. This integration, rather than SLPO alone, enables the strong performance–efficiency trade-offs demonstrated in both in-domain and out-domain tasks (Tables 1–2).
>
> > The symbolic format, though efficient for mathematical reasoning, may not generalize well to broader or open-ended reasoning domains such as commonsense or multimodal tasks, where discrete symbolic operators are harder to define. It's recommended to evaluate the method on more general tasks.
>
> Thank you for this valuable suggestion. Although the model was trained primarily on mathematical reasoning data, we evaluated it on some non-mathematical benchmarks, including GPQA, LSAT, and MMLU, to assess generalization. As shown in Table 2, our models maintain or even improve accuracy on these tasks while achieving up to 15× compression, indicating that the Mentalese-aligned representation transfers effectively to broader reasoning settings even without explicit training on those tasks.
> Extending Mentalese to multimodal or commonsense reasoning, where symbolic operators must handle perceptual or contextual grounding, is indeed an exciting direction for future work, and we plan to explore this in subsequent research.

---

### Note · Program_Chairs · 2026-01-17
**Submission Desk Rejected by Program Chairs**

The following references in this submission do not refer to real documents and/or have major errors in bibliographic information:

 Chain-of-thought reasoning in transformers: A theoretical perspective,